

# Genome-wide identification of gene families related to miRNA biogenesis in *Mangifera indica* L. and their possible role during heat stress

Andrés G. López-Virgen[1], Mitzuko Dautt-Castro[1],
Lourdes K. Ulloa-Llanes[1], Sergio Casas-Flores[2],
Carmen A. Contreras-Vergara[1], Miguel A. Hernández-Oñate[1],
Rogerio R. Sotelo-Mundo[3], Rosabel Vélez-de la Rocha[4] and
Maria A. Islas-Osuna[1]

[1] CTAOV, Centro de Investigación en Alimentación y Desarrollo, A.C., Hermosillo, Sonora, México
[2] División de Biología Molecular, Instituto Potosino de Investigación Científica y Tecnológica, San Luis Potosi, San Luis Potosi, México
[3] CTAOA, Centro de Investigación en Alimentación y Desarrollo, A.C., Hermosillo, Sonora, México
[4] Unidad Culiacán, Centro de Investigación en Alimentación y Desarrollo, A.C., Culiacán, Sinaloa, México

Corresponding authors
Mitzuko Dautt-Castro,
mitzuko.dautt@ciad.mx
Maria A. Islas-Osuna,
islasosu@ciad.mx

## ABSTRACT

Mango is a popular tropical fruit that requires quarantine hot water treatment (QHWT) for postharvest sanitation, which can cause abiotic stress. Plants have various defense mechanisms to cope with stress; miRNAs mainly regulate the expression of these defense responses. Proteins involved in the biogenesis of miRNAs include DICER-like (DCL), ARGONAUTE (AGO), HYPONASTIC LEAVES 1 (HYL1), SERRATE (SE), HUA ENHANCER1 (HEN1), HASTY (HST), and HEAT-SHOCK PROTEIN 90 (HSP90), among others. According to our analysis, the mango genome contains five *DCL*, thirteen *AGO*, six *HYL*, two *SE*, one *HEN1*, one *HST*, and five putative *HSP90* genes. Gene structure prediction and domain identification indicate that sequences contain key domains for their respective gene families, including the RNase III domain in DCL and PAZ and PIWI domains for AGOs. In addition, phylogenetic analysis indicates the formation of clades that include the mango sequences and their respective orthologs in other flowering plant species, supporting the idea these are functional orthologs. The analysis of *cis*-regulatory elements of these genes allowed the identification of MYB, ABRE, GARE, MYC, and MeJA-responsive elements involved in stress responses. Gene expression analysis showed that most genes are induced between 3 to 6 h after QHWT, supporting the early role of miRNAs in stress response. Interestingly, our results suggest that mango rapidly induces the production of miRNAs after heat stress. This research will enable us to investigate further the regulation of gene expression and its effects on commercially cultivated fruits, such as mango, while maintaining sanitary standards.

# INTRODUCTION

The discovery of the role of miRNAs in regulating genes involved in plant homeostasis during stress has changed our understanding of plant physiology (*Ohama et al., 2017*). In crop species, heat stress is a critical limitation for plant growth (*Liu et al., 2017*). Mango is a fruit known for its delicious taste, unique flavor, and various uses. Along with India, China, Thailand, and Indonesia, Mexico is one of the primary producers of this fruit. Due to its growing importance in the global market and its short shelf life, different transcriptomic studies have been conducted on various mango varieties to understand the molecular basis of their responses to varying conditions like development, ripening, and abiotic stress (*Dautt-Castro et al., 2018*; *Tafolla-Arellano et al., 2017*; *Dautt-Castro et al., 2015*). Additionally, sequencing the mango genome has simplified genetic and genomic studies of this species.

The trade of mangoes in commerce can introduce pests like the fruit fly, which is why strict regulations exist. The United States demands a quarantine hot water treatment (QHWT) before importing mangoes to prevent exotic fruit fly larvae from spreading (*USDA-APHIS, 2016*). QHWT consists of submerging the fruit in hot water at 115°F for 60 to 120 min, depending on the fruit size. Besides killing the larvae, the treatment accelerates fruit softening, reducing the shelf life of the fruit. We previously described mRNA seq studies of the effects of QHWT treatment in mango (*Dautt-Castro et al., 2018*).

In recent years, there has been a lot of interest in RNA silencing in plants, a process where small RNA molecules, known as sRNAs, play a crucial role. These sRNAs can vary in length from 21 to 24 nucleotides (nt), and the most common types are microRNAs (miRNAs) and small interfering RNAs (siRNAs). These molecules are essential regulators of various plant functions, such as development, reproduction, defense mechanisms, epigenetic regulation, and maintaining cellular homeostasis (*Borges & Martienssen, 2015*). All types of sRNA biogenesis involve the participation of DICER-LIKE (DCL) and the action of the effector ARGONAUTE (AGO). In the case of miRNAs, these are a unique type of sRNA encoded in microRNA genes (*MIR*), transcribed by RNA polymerase II (Pol II). As a result, a primary transcript (pri-miRNA) with a long stem-loop structure is produced and further cleaved by DCL1 to generate a precursor miRNA (pre-miRNA). The pri-miRNA processing is regulated by microprocessor proteins like SERRATE (SE) and HYPONASTIC LEAVES 1 (HYL1) to undergo a second cleavage by DCL1 to generate a miRNA: miRNA* duplex (*Ivanova et al., 2022*; *Ding & Zhang, 2023*). HASTY (HST) regulates the transcription of *MIR* genes by interacting with RNA polymerase II but also interacts with DCL1 for the correct processing of pri-miRNA (*Cambiagno et al., 2021*; *Bielewicz et al., 2023*). The miRNA duplex is later methylated by HUA ENHANCER1 (HEN1) on its 2′ OH of the 3′-terminal nucleotide, thus protecting against 3′ uridylation and truncation (*Tsai et al., 2014*). The miRNA is then loaded into AGO1, the passenger strand (miRNA*) is degraded, and the miRNA-induced silencing complex (miRISC) is

assembled. This process requires the participation of some cofactors, such as HEAT-SHOCK PROTEIN 90 (HSP90), to help modify AGO1 conformation, opening its nucleic acid-binding channel. The miRISC is transported from the nucleus to the cytoplasm in a CRM1 (EXPO1)/NES-dependent manner *via* TREX-2 and a nucleoporin protein (NUP1) facilitated pathway (*Ding & Zhang, 2023*; *Nakanishi, 2016*). The miRISC binds to cytosolic mRNAs using the complementarity between the miRNA and the target molecule and can act at two levels: transcript cleavage or translational repression (*Iwakawa & Tomari, 2022*).

Many studies have identified various relevant horticultural *DCL* and *AGO* genes in plants. For instance, they have been studied in tomato (*Bai et al., 2012*), Arabidopsis (*Barciszewska-Pacak et al., 2015*), grapevine (*Zhao et al., 2015*), apple (*Xu et al., 2016a*), rice (*Mangrauthia et al., 2017*), cucumber (*Gan et al., 2017*), sweet orange (*Sabbione et al., 2019*), sunflower (*Podder et al., 2023*), and quinoa (*Yun & Zhang, 2023*) to analyze their role under stress conditions (*Cao et al., 2016*; *Gan et al., 2017*). Results from these studies have suggested that the sRNA machinery is highly conserved in plants.

Hence, studying the molecular processes associated with heat stress and gene regulation mechanisms, especially those related to sRNAs, is crucial. This research focuses on identifying and characterizing the genes responsible for miRNA biogenesis machinery, such as *DCL, AGO, HYL1, SE, HEN, HST*, and *HSP90* in the *M. indica* genome. Additionally, we assessed the impact of QHWT on the relative expression of these genes.

## MATERIALS AND METHODS

### Identification of mango *DCL, AGO, HYL1, SE, HEN1, HST,* and *HSP90* gene families and analysis of their sequences

Amino acid (aa) sequences for the DCL, AGO, HYL1, SE, HEN, HST, and HSP90 proteins from *Arabidopsis thaliana* and *Solanum lycopersicum* (tomato) were downloaded from NCBI to search for orthologs in the 'Tommy Atkins' Mango Genome Database (*Bally et al., 2021*) through the BLASTp tool. All sequences of the deduced proteins of interest were obtained from the proteome of *M. indica*. Next, conserved domains of each protein sequence were identified using the Pfam 36.0 server (*El-Gebali et al., 2019*). All candidate sequences were aligned using CLUSTAL W (*Larkin et al., 2007*) and were subjected to further analysis.

Information about the studied genes, such as IDs, chromosomal locations, nucleotide (nt) sequences, and deduced polypeptide sequences, were obtained from the Mango Genome Database (https://mangobase.org). The Compute pI/Mw tool on the Expasy server determined the theoretical isoelectric point (pI) and molecular weight (MW) (*Duvaud et al., 2021*). To obtain the exon-intron structure of the identified genes, full-length nucleotide sequences and their respective coding sequence (CDS) were introduced in the Gene Structure Display Server (GSDS) (*Hu et al., 2015*).

To determine the possible presence of MiHSP90 in the nucleus, three different nuclear localization sequence (NLS) predictors were used: cNLS Mapper (*Kosugi et al., 2009*),
NLStradamus (*Nguyen Ba et al., 2009*), and NucPred (*Brameier, Krings & Maccallum, 2007*). To corroborate the results, the known nuclear-localized HSP90.2 from *Arabidopsis thaliana* was analyzed to make comparisons. cNLS was set to a cut-off score of 6, and searches were performed in all protein regions for a bipartite NLSs with a long linker (13–20 aa). NLStradamus was used in default settings, with a two-state HMM static model and 0.6 prediction cutoff. NucPred was utilized under standard operating conditions without any specialized configurations.

## Phylogenetic analysis of mango genes compared to other plant species

Amino acid sequences of DCL, AGO, HYL1, SE, HEN, HST, and HSP90 proteins from two monocot species (maize and rice), three dicots (Arabidopsis, tomato, and sweet orange), one lycophyte (*Selaginella moellendorffii*), and one bryophyte (*Physcomitrella patens*) were downloaded from the NCBI database (Table S1), and Phytozome v12.1. was used for the analysis. Full-length sequences, including those from mango, were then aligned using the MUSCLE algorithm in default settings clustering by UPGMA (unweighted pair group method with arithmetic mean) (*Edgar, 2004*). Subsequently, phylogenetic trees for each protein set were constructed using the neighbor-joining (NJ) method (*Saitou & Nei, 1987*) through the model Jones-Taylor-Thornton with bootstrap values of 1,500. All analyses were performed in the program MEGA X (*Kumar et al., 2018*).

## Analysis of *cis*-regulatory elements of identified genes

A total of 1.5 kb of nt sequences upstream of the respective start codon (ATG) was extracted from the Mango Genome Database for DCL, AGO, HYL1, SE, HEN, HST, and HSP90 genes. The analysis of *cis*-regulatory elements was performed using the PlantCARE database (https://bioinformatics.psb.ugent.be/webtools/plantcare/html) (*Lescot, 2002*), and the results were visualized and processed using GraphPad Prism 5 software. Stress-related elements were preferentially selected.

## Chromosomal mapping and gene duplication analysis

The chromosomal location of all genes was obtained from the Mango Genome Database. The gene duplication analysis of *DCL*, *AGO*, and *HYL1* families was carried out by calculating the synonymous substitutions (Ks) and non-synonymous substitutions (Ka). This analysis and the illustration of chromosomal location were carried out using the bioinformatic toolkit TBtools (*Chen et al., 2020*).

## Plant material and hydrothermal treatment

Mango (*M. indica* L.) fruit cultivar 'Ataulfo' was harvested at the orchard "La Aviación" located in Escuinapa, Sinaloa, Mexico (22°48′48″N and 105°31 W). Mr. Ismael Díaz Murillo from the "Diazteca" orchard donated the mango fruit used in the experiments. Mango fruits were homogeneous in shape and size, without apparent harm, and at physiological maturity (approximately 120 days after flowering). They were transported to the laboratory at Culiacán, Sinaloa, Mexico (CIAD, AC), disinfected with 200 ppm chlorinated water, and stored at 20 °C for 12 h. QHWT was applied to the fruits (115°F for

75 min), followed by hydrocooling (77°F for 30 min), as established by the *USDA-APHIS (2016)*. Afterward, the untreated and treated mango mesocarp was sampled at 0, 1, 3, 6, and 24 h post-treatment for each condition and immediately frozen in liquid nitrogen, milled, and stored in the freezer. Untreated fruits were considered the controls. Three biological replicates were sampled for treatment and control groups; each replicate comprised three mango fruits.

### RNA extraction and quantitative real-time PCR (qRT-PCR)

Total RNA was isolated from each biological replicate at all time points based on the method described previously by *Lopez-Gomez & Gomez-Lim (1992)* with molecular biology grade reagents (Sigma-Aldrich, Toluca, México) and then treated with RNase-free DNase I (Roche) to eliminate genomic DNA. PCR reactions were made with purified RNA using *MiGAPDH* primers to test for the absence of DNA in the RNA samples. RNA quantity and purity were determined using an Epoch Microplate Spectrophotometer UV-Vis at 260 nm (BioTek, Winooski, VT, USA). RNA integrity was analyzed in 1% agarose gels by electrophoresis under denaturing conditions and visualized with MiniBIS (DNR Bio-Imaging Systems, Neve Yamin, Israel). According to the manufacturer's instructions, first-strand cDNA was synthesized from 2 µg total RNA in a 10 µl total volume reaction using the SuperScript™ III cDNA Synthesis Kit (Invitrogen, Waltham, MA, USA). The resulting cDNAs were diluted to 50 ng/µl and kept at −20 °C.

Gene-specific primers (Table S2) for *MiDCL1, MiDCL3, MiHEN1, MiHST, MiAGO1, MiAGO4, MiAGO6,* and *MiHSP90.3* were designed using Primer Quest Tool and synthesized by IDT (Coralville, IA, USA). *MiGAPDH* was used as a housekeeping gene based on previous studies in mango (*Dautt-Castro et al., 2015*, *2018*). An oligonucleotide concentration of 10 µM was used. qPCR was carried out in a volume of 10 µl, containing two µl of cDNA (100 ng), 0.15 µl of forward primer, 0.15 µl of reverse primer, 5 µl of mix Radiant Green 2X qPCR mix (Thermo Fisher Scientific, Waltham, MA, USA), and distilled water to the final volume. Each cDNA sample was tested in three technical replicates. qPCR reactions were run in a PikoReal Real-Time PCR System (Thermo Fisher Scientific, Waltham, MA, USA) with the following conditions: 95 °C for 7 min, then 40 cycles at 95 °C for 5 s and 60 °C for 30 s. The dissociation curve was from 60–95 °C, with a gradual increase of temperature every 30 s. Relative gene expression levels were calculated using the $2^{-\Delta\Delta Ct}$ method (*Schmittgen & Livak, 2008*). The One-Way ANOVA test was used to analyze relative expression data in NCSS 12 software. The Tukey-Kramer *post hoc* test was used to perform a multiple means comparison with a $p < 0.05$.

## RESULTS

### Identification of miRNA biogenesis-related genes

*A. thaliana* and tomato protein sequences from NCBI were analyzed to identify all *DCL, AGO, HYL1, SE, HEN, HST,* and *HSP90* genes and their deduced protein sequences was used to search the annotated orthologs in the Mango Genome Database through the BLASTp tool. This approach identified five *MiDCL* genes, thirteen *MiAGO* genes, six *MiHYL1* genes, two *MiSE* genes, one gene for *MiHEN1*, one gene for *MiHST*, and five

*MiHSP90* genes (Table 1). All genes were named based on their homology and phylogenetic relationship with proteins from the other organisms analyzed.

Each gene family showed specific characteristics, such as that the MiDCLs and MiHSTs had the highest average sequence length (ORF: 5,107 and 5,160 nt; AA: 1,701 and 1,719 aa; 196.57 and 190.7 kDa, respectively), while MiHYL1 was the smallest family, with 1,090 nt, 363 aa, and 39.33 kDa on average. For the predicted pI, the MiAGO family showed the highest value, 9.28 on average, and the HSP90 family had the smallest average value, 5.06 (Table 1).

To deepen the analysis of the sequences, we determined the structure of each of those genes using GSDS 2.0, based on the intron and exon sequences. The structure of *DCL* genes showed an intron number ranging from 19 to 27 (*MiDCL1* and *MiDCL2b*, respectively) (Fig. S1A). Seven *AGO* genes have at least 20 introns, whereas *MiAGO2* and *MiAGO7* have only two (Fig. S1B). *HYL1* genes had between 1 (*MiHYL1a* and *c*) and 5 (*MiHYL1f*) introns, while *SE* genes both had 11 introns (Figs. S1C and S1D). *MiHEN1* and *MiHST* have 11 and 26 introns, respectively (Figs. S1E and S1F), while the *HSP90* family ranged from 2 (*MiHSP90.3 and 0.4*) to 5 (*MiHSP90.5*) (Fig. S1G). For this family of *HSP90*, hereafter, only *MiHSP90.3* was selected for further analysis, and we refer to this gene as *MiHSP90*. This protein was chosen because of its highly probability of being found in the nucleus, like its homolog in Arabidopsis, so we predict it participates in nuclear miRNA loading into AGO1. To confirm this hypothesis, an analysis of nuclear localization was carried out. cNLS Mapper showed scores above six for both sequences. The MiHSP90.3 sequence found was FVKGIVDSEDLPLNISRETLQQNKILKVI (Fig. S2A). According to the algorithm, the obtained score may suggest that our sequence might be partially localized to the nucleus or localized to both the nucleus and the cytoplasm. NLStradamus found a short KEEKKKKKIK amino acid sequence located within 244–253 residues (Fig. S2B). Simultaneously, a NucPred score of 0.51 was observed, indicating that 70% of proteins predicted as nuclear were indeed nuclear, with corresponding coverage of 62% representing the proportion of true nuclear proteins accurately predicted (Fig. S2C).

## Conserved domain analysis

Deduced amino acid sequences were analyzed using Pfam 36.0 (*El-Gebali et al., 2019*). These analyses revealed that except for MiDCL2a, all MiDCL proteins contained the Helicase-C motif, involved in the unwinding of RNA duplexes during RNA processing steps; the Dicer dimer, which promotes the essential dimerization of the protein allowing efficient processing of dsRNA substrates; the PAZ domain, and at least one Ribonuclease III (RNase III) domain which participates in binding and cleaving double-stranded RNA. The presence of these domains supports the function of these putative sequences as viable DCL proteins. MiDCL1 is the only one that contains a DEAD-box type RNA helicase domain at the N-terminus. In contrast, MiDCL2a and MiDCL2b showed unique PCRF and RF-1 domains, acting as peptide chain release factors. A ResIII domain with endonuclease activity was only identified in MiDCL3 and MiDCL4. In contrast, the D1 domain was found only in MiDCL1 and MiDCL4 (Fig. 1A). Multiple sequence analyses of the DCL domains in mango (Mi), Arabidopsis (At), and tomato (Sl) are shown in Fig. S3.

**Table 1 Characteristics of sRNA biogenesis genes from *Mangifera indica* L.**

| Gene | Sequence ID | Chr | ORF (bp) | Protein | | |
|------|-------------|-----|----------|---------|---|---|
| | | | | AA | Ip | MW (kDa) |
| DICER-LIKE | | | | | | |
| **MiDCL1** | Manin16g013400.1 | 16 | 5,964 | 1,987 | 5.99 | 223.8 |
| MiDCL2a | Manin03g001700.1 | 3 | 4,533 | 1,510 | 5.85 | 169.8 |
| MiDCL2b | Manin03g001710.1 | 3 | 5,094 | 1,697 | 5.93 | 191.4 |
| **MiDCL3** | Manin05g003940.1 | 5 | 5,055 | 1,684 | 6.76 | 188.3 |
| MiDCL4 | Manin05g000540.1 | 5 | 4,887 | 1,628 | 6.43 | 182.8 |
| ARGONAUTE | | | | | | |
| MiAGO1a | Manin00g009890.1 | 0 | 3,669 | 1,222 | 9.64 | 135.2 |
| **MiAGO1b** | Manin17g007920.1 | 17 | 3,294 | 1,097 | 9.39 | 121.8 |
| MiAGO2a | Manin13g011150.1 | 13 | 2,955 | 984 | 9.47 | 109.4 |
| MiAGO2b | Manin18g011630.1 | 18 | 3,045 | 1,014 | 9.32 | 113.1 |
| **MiAGO4a** | Manin01g005840.1 | 1 | 2,661 | 886 | 8.89 | 99.1 |
| MiAGO4b | Manin03g006020.1 | 3 | 1,647 | 548 | 8.97 | 61.3 |
| MiAGO5 | Manin03g010820.1 | 3 | 2,943 | 980 | 9.6 | 109.6 |
| **MiAGO6** | Manin12g007340.1 | 12 | 2,760 | 919 | 9.12 | 103.2 |
| MiAGO7a | Manin09g012380.1 | 9 | 3,039 | 1,012 | 9.13 | 115.3 |
| MiAGO7b | Manin16g006430.1 | 16 | 3,036 | 1,011 | 9.27 | 115.0 |
| MiAGO10 | Manin02g009270.1 | 2 | 2,979 | 992 | 9.23 | 111.5 |
| MiAGOMEL1 | Manin16g005400.1 | 16 | 2,784 | 927 | 9.47 | 103.8 |
| MiAGOPNH1 | Manin07g002280.1 | 7 | 2,757 | 918 | 9.24 | 104.5 |
| HEN | | | | | | |
| **MiHEN1** | Manin18g010850.1 | 18 | 3,630 | 1,209 | 6.20 | 135.3 |
| HASTY | | | | | | |
| **MiHST** | Manin03g011860.1 | 3 | 5,160 | 1,719 | 6.91 | 190.7 |
| HYL1 | | | | | | |
| MiHYL1a | Manin01g011660.1 | 1 | 1,149 | 382 | 7.01 | 41.0 |
| MiHYL1b | Manin14g005250.1 | 14 | 1,185 | 394 | 8.72 | 42.6 |
| MiHYL1c | Manin15g005480.1 | 15 | 1,149 | 382 | 8.09 | 41.0 |
| MiHYL1d | Manin15g010550.1 | 15 | 1,149 | 382 | 8.38 | 41.2 |
| MiHYL1e | Manin16g001590.1 | 16 | 879 | 292 | 7.52 | 32.5 |
| MiHYL1f | Manin18g009710.1 | 18 | 1,029 | 342 | 6.21 | 37.7 |
| SE | | | | | | |
| MiSE1 | Manin01g005900.1 | 1 | 2,289 | 762 | 6.52 | 85.9 |
| MiSE2 | Manin03g005950.1 | 3 | 2,286 | 761 | 8.53 | 85.5 |
| HSP90 | | | | | | |
| MiHSP90.1 | Manin01g011860.1 | 1 | 2,097 | 698 | 4.97 | 80.2 |
| MiHSP90.2 | Manin04g001850.1 | 4 | 2,109 | 702 | 5.02 | 80.8 |
| **MiHSP90.3** | Manin10g008510.1 | 10 | 2,100 | 699 | 5.03 | 80 |
| MiHSP90.4 | Manin19g016320.1 | 19 | 2,103 | 700 | 5.01 | 80.1 |
| MiHSP90.5 | Manin20g008060.1 | 20 | 2,034 | 677 | 5.29 | 77.2 |

**Note:**
Gene expression assays were conducted on the highlighted genes. Chr, Chromosome; AA, Amino acid number; Ip, Isoelectric point; MW, Molecular weight.

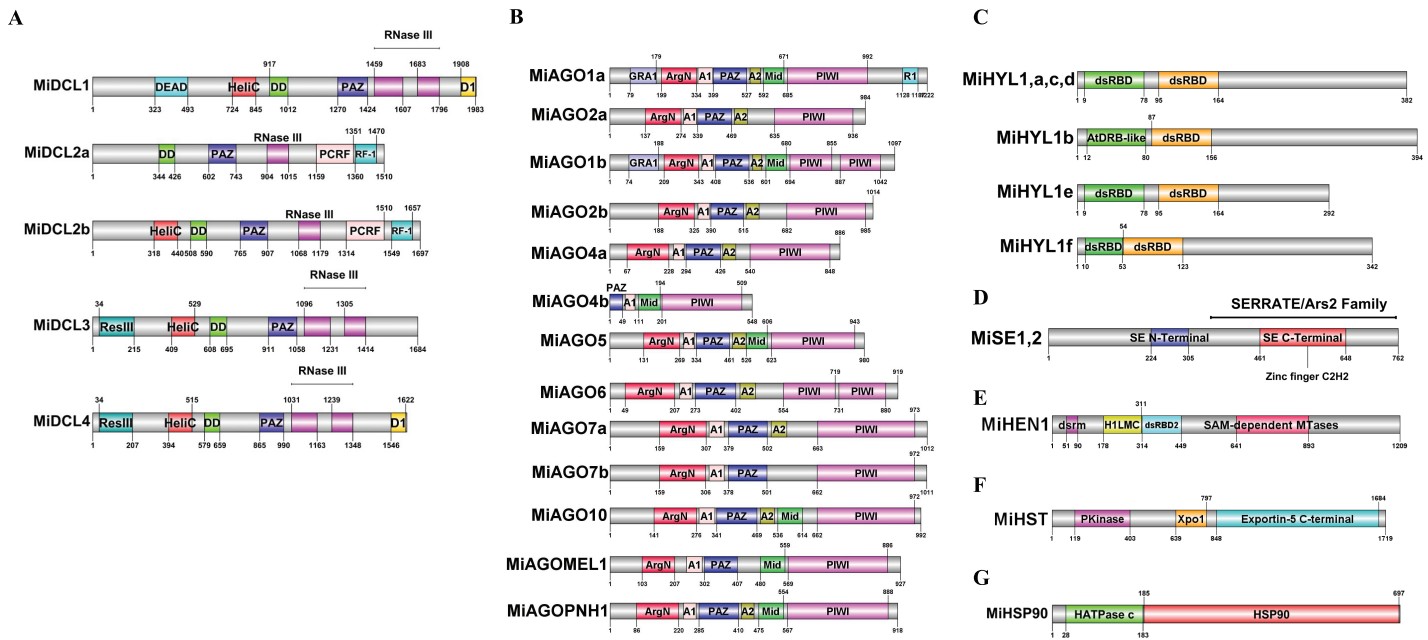

**Figure 1  Domains identified in the deduced amino acid sequences of the DCL (A), AGO (B), HYL1 (C), SE (D), HEN (E), HST (F), and HSP90 (G) proteins.** Domain abbreviations: HeliC, Helicase C; DD, Dicer dimer; RNase III, Ribonuclease III; D1, DND1 DSRM; ResIII, Type III restriction enzyme; RF-1, Release factor; GRA1, Gly-rich AGO1; ArgN, ArgoN; A1, ArgoL1; A2, ArgoL2; Mid, ArgoMid; R1, RRM1; dsRBD2, Double-stranded RNA binding domain 2; H1LC, HEN1 Lam C; Pkinase, Protein kinase; Xpo1, Exportin 1; HATPase, ATPase binding.

Except for MiAGO4b, all putative AGOs contain the N-terminal ArgoN domain, which is linked to the PAZ domain *via* ArgoL1 (A1) (*Yuan et al., 2005*; Fig. 1B); both are present in all 13 proteins. ArgoL2 (A2) (Fig. 1B) also helped connect these domains and was found in 12 of 13 proteins. The Mid domain (Fig. 1B), found in 7 AGOs, binds small RNAs at the 5′ end. Meanwhile, the PAZ domain binds at the 3′ end. This fact helps the PIWI domain, also present in the 13 AGO genes of mango, to cleave the target accurately through its endonucleolytic activity (*Höck & Meister, 2008*). In contrast, MiAGO1a and MiAGO1b include a Gly-rich AGO1 domain (GRA1) (Fig. 1B), which coordinates binding with the ribosome to enhance AGO protein stimulation for RNA silencing. Additionally, MiAGO1a has an RRM1 (R1) (Fig. 1B) domain that is absent in the rest of the mango AGOs. MiAGO1b and MiAGO6 are also the only ARGONAUTES with two PIWI domains in their structure. Multiple sequence analyses of the AGO domains in mango (Mi), Arabidopsis (At), and tomato (Sl) are in Fig. S4.

For the HYL1 family, the double-stranded RNA-binding domain (dsRBD) at the N-terminus, which is essential for miRNA processing, was found twice in each protein, as it has been reported in other plants (*Wu et al., 2007*; Fig. 1C). Multiple sequence analysis of the HYL1 domains in mango (Mi), Arabidopsis (At) and tomato (Sl) are in Fig. S5. Both mango SE proteins have a zing-finger domain at the N-terminal domain, and the C-terminal domain, and this is reported in Arabidopsis to bind pri-miRNA (*Machida, Chen & Adam Yuan, 2011*; Fig. 1D). Multiple sequence analyses of the SE domains in

mango (Mi), Physcomitrella *patens* (Pp), *Selaginella moellendorffii* (Sm), *Oryza sativa* (Os), *Cucumis sativus* (Cs), *Zea mays* (Zm), Arabidopsis (At) and tomato (Sl) are in Fig. S6.

In addition, for MiHEN1, the H1LC, dsRBD2, and 2'-O-methyltransferase Hen1 domains were found, which are related to its role as RNA methyltransferase (Fig. 1E). Multiple sequence analysis of the HEN1 domains in mango (Mi), Arabidopsis (At) and tomato (Sl) are in Fig. S7. The protein structure of MiHST contains a P kinase domain, Xpo1 which is involved in the translocation of proteins out of the nucleus, and Exportin-5 (*Yi et al., 2003*; Fig. 1F). Multiple sequence analysis of the HST domains in mango (Mi), Arabidopsis (At) and tomato (Sl) are in Fig. S8. Finally, MiHSP90 includes the histidine kinase-like ATPase (HATPase), and a domain identified as HSP90, probably corresponding to the characteristic binding to substrate region in this family of proteins (*Makhnev & Houry, 2012*; Fig. 1G). Multiple sequence analyses of the HSP90 domains in mango (Mi), Arabidopsis (At) and tomato (Sl) are in Fig. S9.

## Phylogenetic analysis of DCL, AGO, HYL1, SE, HEN1, HST, and HSP90 proteins

All full-length protein sequences from diverse plant species, including mango, maize, tomato, sweet orange, rice, *A. thaliana*, *S. moellendorffii*, and *P. patens* were aligned and used to construct a neighbor-joining (NJ) tree to infer the phylogenetic relationships among the DCL, AGO, HYL1, SE, HEN, HST, and HSP90 proteins. The DICER-like tree consists of four main clades, representing the main types of DCL in plants (DCL1, DCL2, DCL3, and DCL4) (*Liu, Feng & Zhu, 2009*; Fig. 2A). The AGO tree arrangement was comparable with the reported phylogenetic tree by *Zhang et al. (2015)*. This tree is confirmed by three main clades grouping the AGO1/5/10, AGO2/3/7, and AGO4/6/8/9/ 16. Additionally, AGOPNH1 and AGOMEL1, which are orthologous identified in rice (*Nishimura et al., 2002*; *Nonomura et al., 2007*), were contained in the AGO1/5/10 clade (Fig. 2B). In the HYL1 case, all mango proteins were grouped and shared a clade with *C. sinensis*, similar to what occurred with SE proteins (Figs. 2C and 2D). For HEN, two clades identified include a single sequence (*S. moellendorffii* and *P. patens*). At the same time, *O. sativa* and *Z. mays* are grouped in the same clade, while the rest of the species align to the other node, where the mango sequence is located (Fig. 2E). In the HST tree, the structure is very similar to the HEN phylogeny; however, *S. moellendorffii* and *P. patens* are in the same clade, resulting in three different central nodes (Fig. 2F). Interestingly, the HSP90 tree showed a different structure, where three main nodes are displayed, one including a single sequence *(O. sativa)*, a second clade including two sequences (*A. thaliana* and *S. lycopersicum)*, and a third group of sequences where mango protein consistently align with the proteins from orange (Fig 2G). In all built trees, the sequences belonging to *M. indica* were closely related to *C. sinensis*.

## Analysis of *cis*-regulatory elements

To analyze the potential regulatory elements in the promoters of each member of the seven gene families, 1.5 Kb upstream of the start codon of each gene was analyzed. The analysis

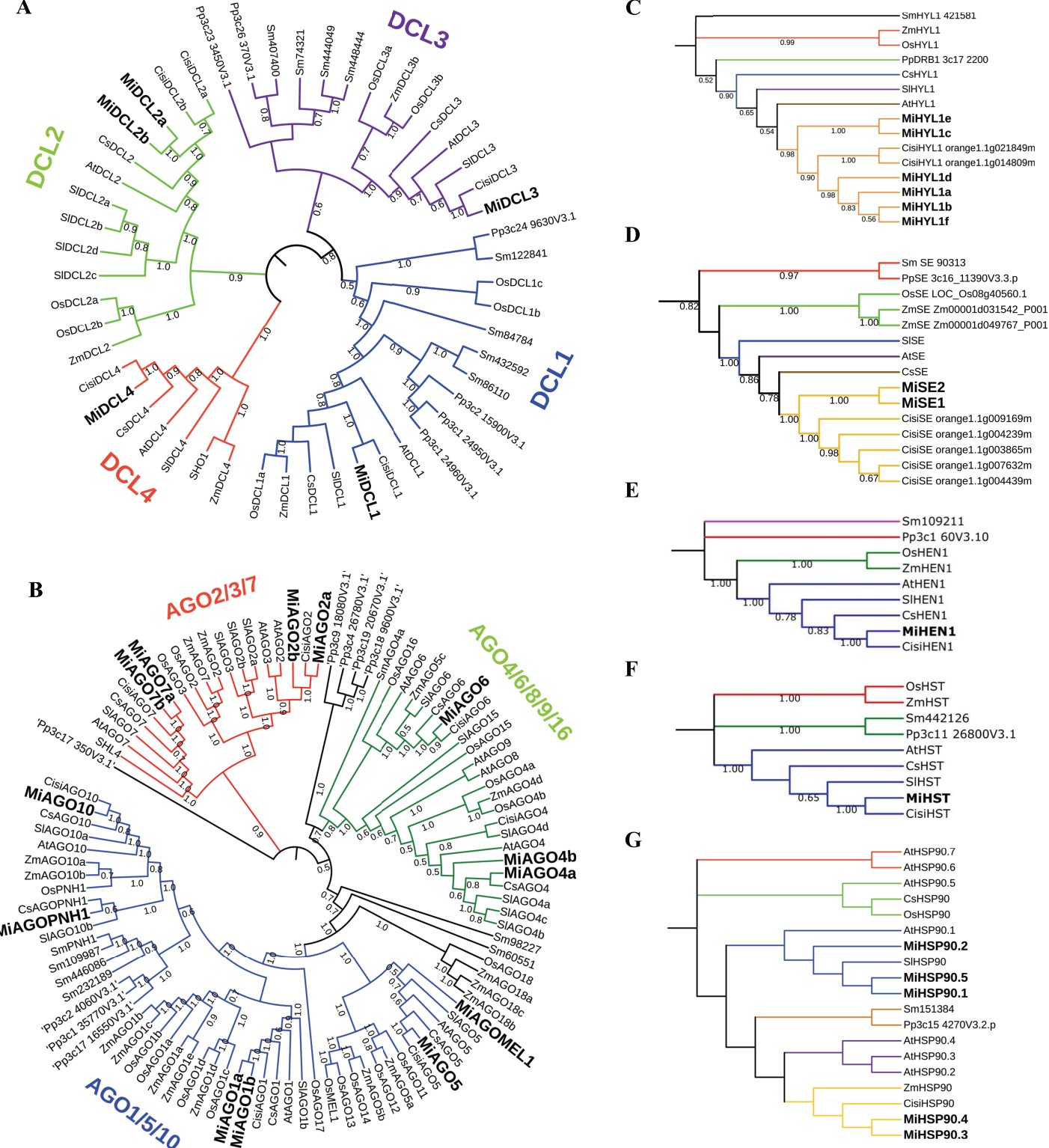

**Figure 2 Phylogenetic trees of the sequences identified in mango with respect to other plant species.** Inferences from MiDCL, MiAGO, MiHYL1, MiSE, MiHEN1, MiHST, and MiHSP90. Sequence abbreviations are listed below: Mi (*Mangifera indica*), Cisi (*Citrus sinensis*), At (*Arabidopsis thaliana*), Sl (*Solanum lycopersicum*), Cs (*Cucumis sativus*), Os (*Oryza sativa*), ZM (*Zea mays*), Sm (*Selaginella moellendorffii*) and Pp (*Physcomitrella patens*). The number in each branch indicates the bootstrap value of the inference.

showed the promoters contain many *cis*-elements related to heat, drought, and hypoxia stress responses. The top 3 *cis*-elements found are MYB, with 135 elements distributed in 26 of 29 sequences, MeJA-responsive with 69 elements distributed in 16 sequences, and MYC, with 65 elements distributed in 25 of 29 promoter regions (Fig. 3). ERE, known as an ethylene-responsive element, and ARE, a *cis*-acting regulatory element essential for anaerobic induction, were present in 23 and 21 of all sequences analyzed, respectively (Fig. 3). Other important *cis*-regulatory elements found are ABRE and GARE (Table S3). These results suggest that most of the miRNA machinery genes have stress-responsive elements related to heat and hypoxia, which are relevant to the conditions studied in the QHWT.

## Chromosomal mapping and gene duplication analysis

Figure 4 shows the chromosomal position of the identified *MiDCL*, *MiAGO*, *MiHYL1*, MiSE, *MiHEN*, *MiHST*, and *MiHSP90*. These genes are distributed in 15 of the 20 chromosomes. Chr 3 has the most, with two *MiDCL*, two *MiAGO*, one *MiHST*, and one *MiSE*.

The duplication events demonstrate the expansion of the gene families in mango fruit. Therefore, we identified gene pairs and types of duplication events in the *MiAGO*, *MiDCL*, and *MiHYL1* families (Figs. 4 and 5). A total of six gene pairs were identified in the *MiAGO* gene family. All these gene pairs are in different chromosomes, suggesting that segmental duplication is the primary expansion model of the *MiAGO* family. On the other hand, both gene pairs identified in the *MiDCL* gene family are on the same chromosome, with one pair organized as a tandem duplication, which suggests that local duplication propelled some of the expansion of the *MiDCL* family (Fig. 4). For the *MiHYL1* family, both kinds of duplication events were found. *MiHYL1e* and *f* appear to be products of segmental duplication (Fig. 5), while *MiHYL1c* and *d* were duplicated in tandem (Fig. 4).

Additionally, we estimated the divergence time of the obtained *AGO* gene pairs. Most of the analyzed gene pairs diverged after the reported ancestral polyploidization (65 million years ago) of the mango fruit genome, a shared event with *Anacardium occidentale* (Table S4; *Bally et al., 2021*). Interestingly, the following gene pairs, MiAGO6-MiAGO4a, MiAGOPNH1-MiAGO10, and MiDCL4-MiDCL3, diverged before the ancestral polyploidization event mentioned above (Table S4).

Furthermore, using synonymous and nonsynonymous substitutions, the Ka/Ks ratio was calculated for the gene pairs in the three families. This aims to understand the evolutionary constraints that acted on *MiAGO* and *MiDCL* gene families. This analysis demonstrated that except for *MiHYLe* and *f,* the rest of the orthologous gene pairs evaluated exhibited Ka/Ks < 1, meaning that *MiAGO* and *MiDCL* have undergone purifying or negative selective pressures during evolution. Those of *MiHYL1* with 1.25 Ka/KS *ratio* have undergone positive selection (*Wen et al., 2020*).

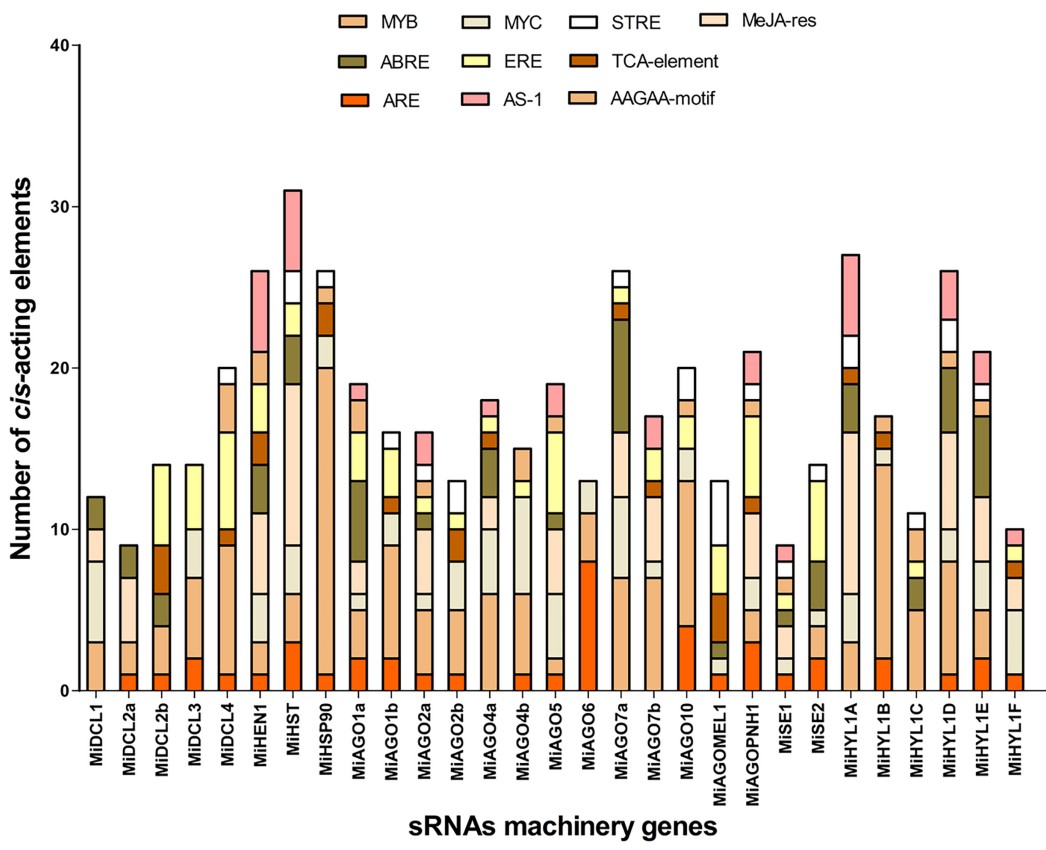

**Figure 3** **Top 10 *Cis*-acting regulatory elements in the upstream promoter region of *MiDCL, MiAGO, MiHYL1, MiSE, MiHEN1, MiHST,* and *MiHSP90* genes.** Different colors represent the binding site for each regulatory element. MYB, MYB-type transcription factor; ABRE, ACGT containing ABA response elements; ARE, *cis*-acting regulatory element for anaerobic induction; MYC, MYC-type transcription factor; ERE, ethylene-responsive element; AS-1, activated by salicylic acid; STRE, Stress-Mediated *cis*-Element Transcription Factor; TCA-element; AAGAA-motif; MeJa-res, methyl jasmonate responsiveness.

## Expression analysis of studied genes in response to hydrothermal treatment

Two *DCLs* (*MiDCL1* and *MiDCL3*), three *AGOs* (*MiAGO1b*, *MiAGO4a*, and *MiAGO6*), and one *MiHEN1, MiHST,* and *MiHSP90* genes were chosen for expression analysis in mango mesocarp through time after QHWT application. These genes were selected based on previous studies and literature supporting their roles in abiotic stress. All analyzed genes showed similar expression patterns in control fruit through time, where a slight induction was found between 1- and 3-h post-treatment up to a two-fold maximum. All genes showed different expression patterns induced by the QHWT in fruit mesocarp. *MiAGO4* and *MiHST* showed an up-regulation starting 1 h after the QHWT, although their maximum expression levels were reached at 24 and 3 h, respectively. At 3 h after treatment, the *MiDCL1, MiAGO1, MiHEN1,* and *MiHSP90* genes responded, increasing their expression levels. At the same time, *MiDCL3* and *MiAGO6* were up-regulated after 6 h post-treatment (Fig. 6). These differential modulations in the transcript levels indicate

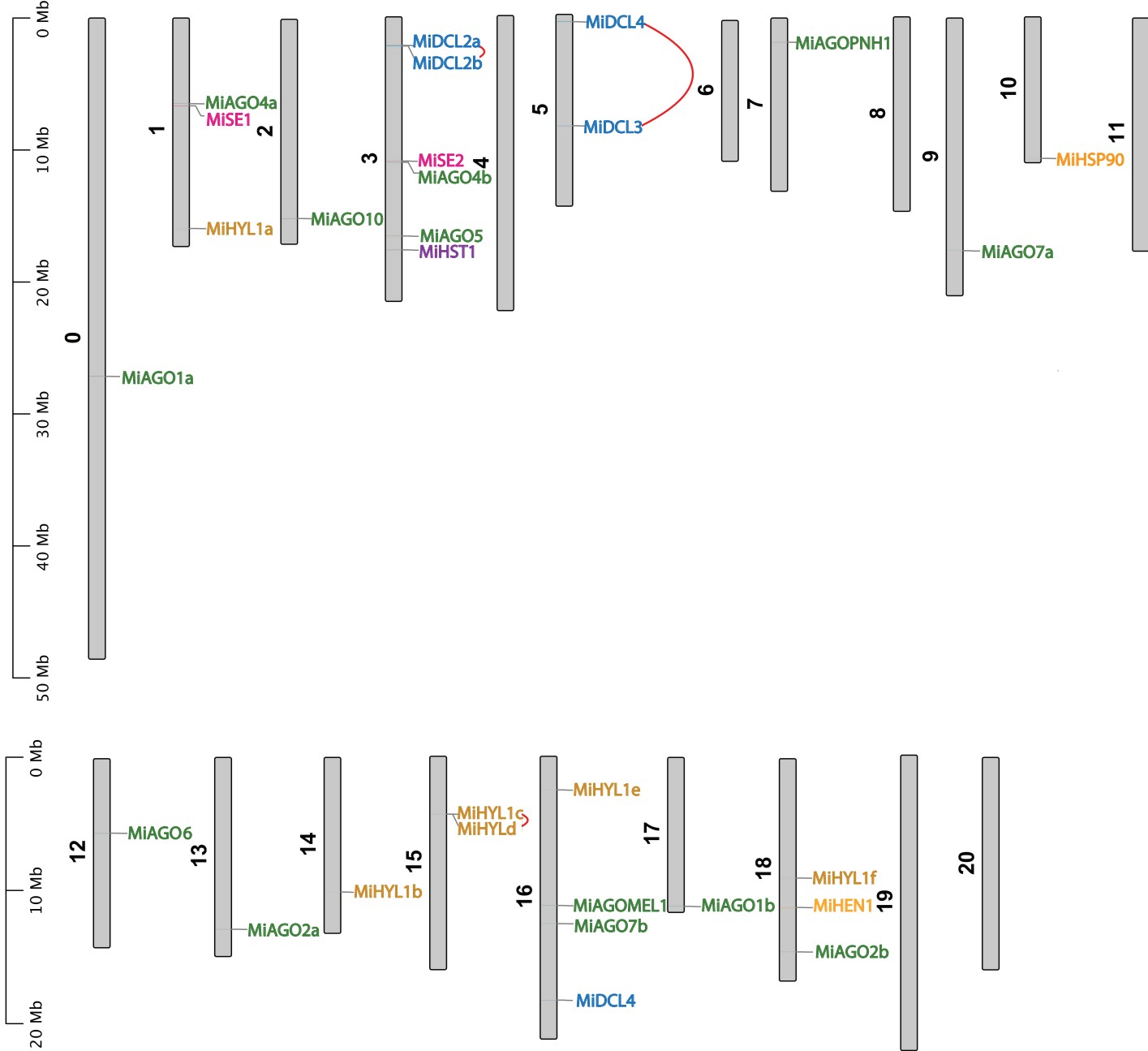

**Figure 4  Chromosomal localization of *DCL*, *AGO*, *SE*, *HYL1*, *HEN1*, *HST*, and *HSP90*.** Each gene position is indicated according to the mango genome. Tandemly duplicated *DCL* gene pairs are connected by red lines.

that the elements of miRNA-biogenesis machinery evaluated here respond to heat stress, increasing the expression at early times after 1 or 3 h post-treatment and this maintained until 6 h. Furthermore, these results could suggest a specific regulation of the different sRNA biogenesis pathways in response to QHWT.

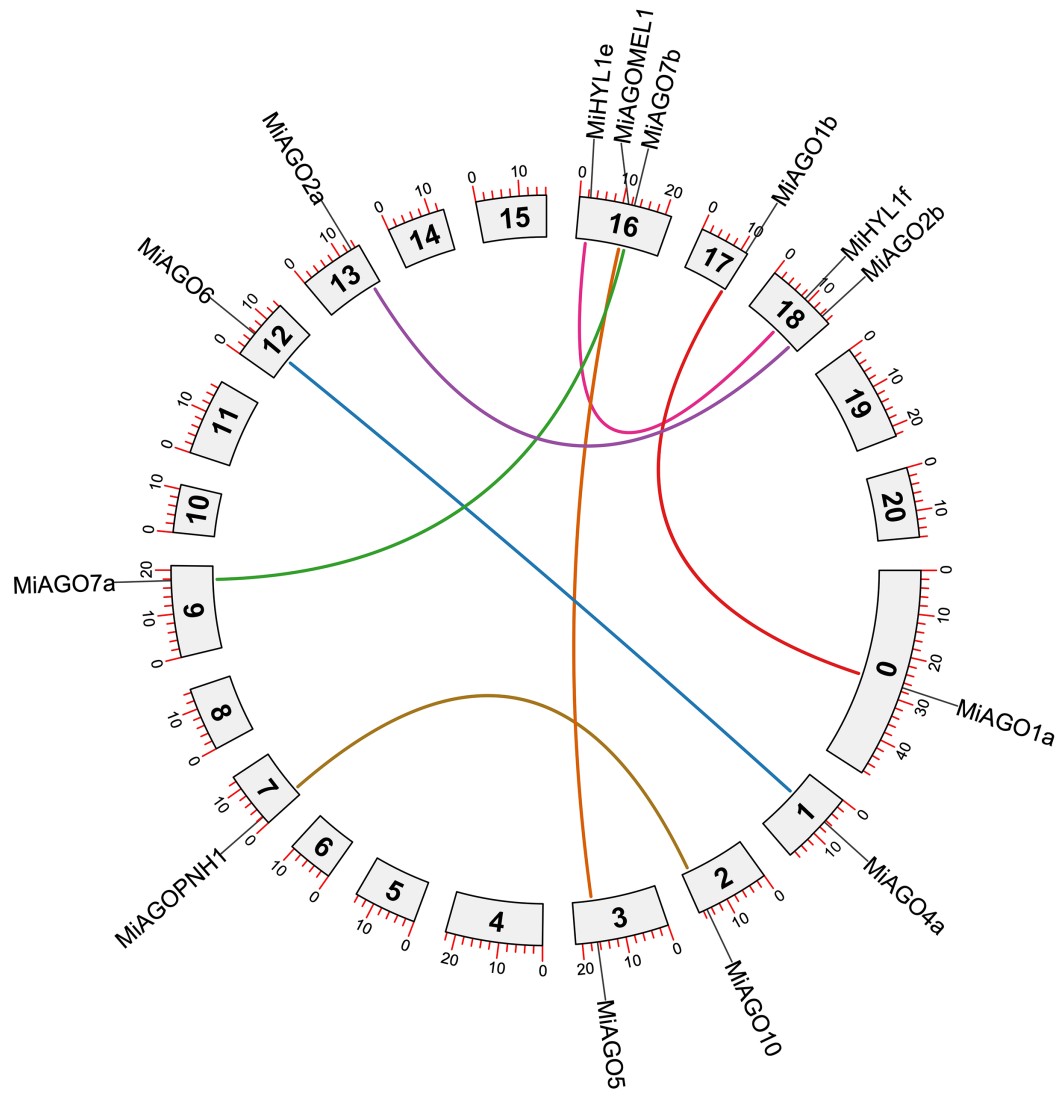

**Figure 5 Circos plot representing the segmental duplicated *AGO* and HYL1 gene pairs.** Each duplication pair is represented with different colors.

## DISCUSSION

Small RNAs are one of the most important mechanisms in plants for regulating gene expression. The recognition of miRNAs as essential regulators in plant development and key molecules for improving agricultural traits has made identifying the proteins involved in their biogenesis crucial. Thus, genes involved in these processes have been identified in many plant species, such as maize (*Qian et al., 2011*), tomato (*Bai et al., 2012*), grapevine (*Zhao et al., 2015*), cucumber (*Gan et al., 2016*), and pepper (*Qin et al., 2018*), among other crop species. However, those gene products involved in miRNA biogenesis remain to be identified in *M. indica*. In the present investigation, the mango genome and transcriptomes allowed the identification of the corresponding genes encoding for *DCL, AGO, HYL1, SE, HEN, HST*, and *HSP90*.

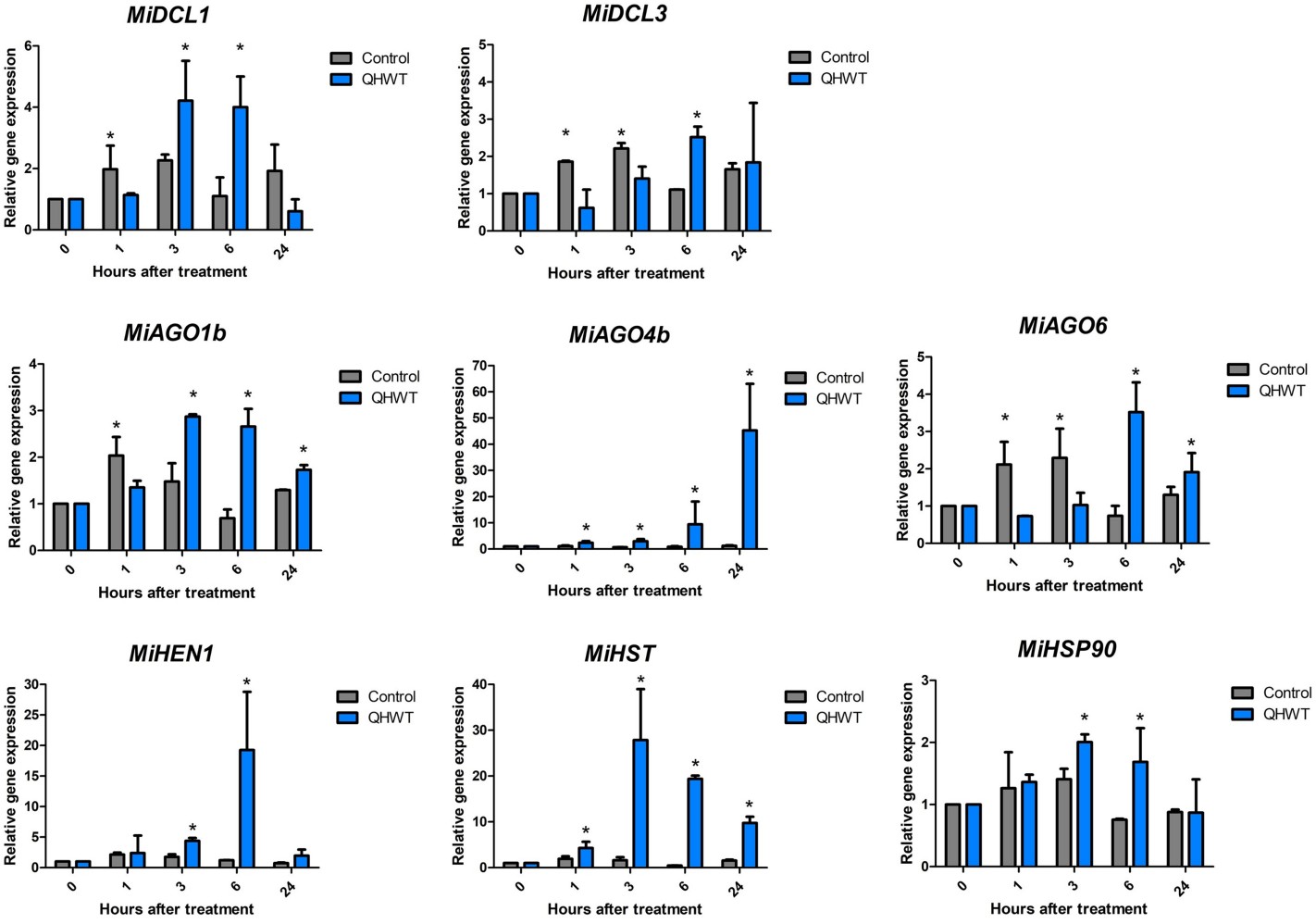

**Figure 6 Expression levels of the *DCL, AGO, HEN, HST*, and *HSP90* selected genes.** The values shown indicate the relative expression of the mango cv. Ataulfo genes subjected to QHWT and without QHWT (control) ($n = 3$ with three experimental replicates each). Data were analyzed using the one-way ANOVA test. Asterisks (*) indicate significant statistical differences between the control and QHWT according to t-test analysis ($p < 0.05$).

Here, we show that the mango genome contains the primary gene families associated with miRNA biogenesis, and they exhibit a response to heat stress triggered by QHWT application to fruits. Similar gene numbers have been reported in other plant species. We identified five *DCL* genes in the mango genome, while in *A. thaliana*, four have been reported (*Bologna & Voinnet, 2014*), eight in rice (*Kapoor et al., 2008*), seven in tomato (*Bai et al., 2012*), four in grapevine (*Zhao et al., 2015*), and five in cucumber (*Gan et al., 2016*). For *AGO* genes, these numbers are at least 10, 19, 15, 13, and 7 for the species mentioned above, while in mango, we found 13. Regarding HYL1, these are part of the double-stranded RNA (dsRNA) binding domain proteins (dsRBP or DRB). *A. thaliana* genome encodes *DRB1* to *DRB5*, where *DRB1* is homologous of *HYL1*, which is what DCL1 requires it for miRNA biogenesis (*Hiraguri et al., 2005*; *Szarzynska et al., 2009*). We found six genes coding for *HYL1* in mango, while Arabidopsis only has a single copy of this gene (*Han et al., 2004*). These could be explained by the gene duplication observed in

two *HYL1* gene pairs in mangoes. On the other hand, also a single copy of *SE* is encoded in Arabidopsis (*Yang et al., 2006*), one in tomato, three in pepper (*Capsicum annuum*) (*Mueller et al., 2005*), and five in orange (*Wu et al., 2014*), compared with two in mango. Although *HEN1* and *HST* have not been widely studied compared to other molecules in the pathway such as *AGO* and *DCL*, findings in *A. thaliana* elucidated one gene from each of these families (*Park et al., 2005*; *Tsai et al., 2014*), just as we found in mango.

## DICER and ARGONAUTE proteins in mango

DCL and AGO have been the most studied gene families. DICER-like proteins are endoribonuclease (RNase) III enzymes that produce siRNA from longer double-stranded RNAs (dsRNAs) and miRNA from single-stranded RNAs with internal stem-loop structures (*Fukudome & Fukuhara, 2017*). AGO proteins are core molecules involved in RNA silencing. In plants, the roles of AGO have been ascribed to regulatory mechanisms at the post-transcriptional and transcriptional gene silencing level (*Fang & Qi, 2016*). Here, we found that all MiDCLs are in the same clade as the DCLs from *C. sinensis*, such as what occurred for MiAGO proteins, suggesting that these species are closely related, as we previously reported (*Dautt-Castro et al., 2015*, *2018*). The other four families also presented this close relationship with *C. sinensis*.

Moreover, some mango DCL sequences were included in the same nodes as other fruit plants, such as tomatoes from different branches of dicots and other groups of plants. These results are similar to those found in analogous studies with a grapevine (*Zhao et al., 2015*). Furthermore, some mango *AGO* genes clustered along with other fruit plants, such as tomato and cucumber, and in different clades than grasses like maize and rice, which even have their lineage-specific *AGO* (*Kapoor et al., 2008*; *Qian et al., 2011*). Also, research has supported the existence of three major clades of AGO proteins in angiosperms: AGO1/5/10, AGO2/3/7, and AGO4/6/8/9 (*Li et al., 2022*). According to our phylogenetic tree, we found a classification comparable to those reported in the literature. As other studies indicate, these results support the divergence between sequences among plant groups (*Zhao et al., 2015*). Additionally, sequences from *P. patens* (bryophyte) and *S. moellendorffii* (lycophyte) contributed to reducing potential biases and testing our hypotheses about the relationships among the studied taxa.

The protein domain search identified conserved domains in the studied proteins. This analysis revealed characteristic domains previously found in tomatoes, grapevines, and cucumbers (*Bai et al., 2012*; *Zhao et al., 2015*; *Gan et al., 2016*). According to other authors, the catalytic core of DICER proteins consists of the PAZ domain, which acts as an anchor for the 3′-end of the dsRNA, and two RNase III domains that catalyze the hydrolysis of a phosphodiester bond within each strand of the dsRNA (*Bernstein et al., 2001*; *Park et al., 2011*). The PAZ domain contains a conserved pocket that recognizes the dsRNA substrate. In contrast, each of the two Ribonuclease III domains is responsible for strand cuts, so finding both in our mango sequences strongly supports the possibility they are functional (*Borges & Martienssen, 2015*). In terms of our findings, identifying the HeliC domain at the N-terminus suggests the presence of the Helicase domain in mango DCL. Moreover, the DEAD domain identified in MiDCL1 corresponds to the basic structure of Arabidopsis

DCL1, which includes the DexD/H-box typical of helicases required to process some pri-miRNAs accurately (*Liu, Axtell & Fedoroff, 2012*). Intriguingly, *Jing et al. (2023)* found this DEAD domain in all DICER proteins analyzed in *Fragaria ananassa* and *F. vesca*, not only in DCL1.

Interestingly, MiDCL2a and MiDCL2b lack one Ribonuclease III domain, and the Helicase C domain is missing in MiDCL2a. Instead, these paralogs only contain the PCRF and RF-1 domains in their C-terminal structure, both related to the translational termination Campo domain (*Song et al., 2000*). In addition to these similarities, these proteins are comparable in their gene structure and genomic sequence length. These results correlate with DCL2a and DCL2b being products of a tandem duplication in Chr 3. The DND1-dsRM domain, homologous to double-strand RNA binding domains, was only found in MiDCL1 and MiDCL4. This finding has been reported for DCL1 in other organisms (*Cardoso et al., 2018*; *Hajieghrari, Farrokhi & Kamalizadeh, 2022*), but not for DCL4, as we saw in mango. Finally, MiDCL3 and MiDCL4 were the only two proteins showing the ResIII domain, characteristic of the DexH/D helicases (*Byrd & Raney, 2012*). Interestingly, our analysis shows these proteins result from tandem duplication in Chr 5.

Regarding the AGO protein's structure, most of the identified ARGONAUTES in mango share the AGO N-terminal domain, AGO linker domain 1 and 2, PAZ, MID, and PIWI domains. These results suggest that these proteins may be functional as they contain all the functional domains. Interestingly, AGO1a and AGO1b were the only sequences showing the presence of the GRA1 domain, reported previously in maize (*Xu et al., 2016b*). Although the authors did not discuss the role of the domain, AGO-associated proteins often contain glycine-tryptophan-rich regions, which are believed to interact with the ARGONAUTE PIWI domain (*Schirle & MacRae, 2012*). MiAGO1b and MiAGO6 were the only two sequences that included two PIWI domains. However, even when a recent study indicated that the PIWI domain could be split into two subdomains (one of them containing an RNase H fold, while the other was composed of helices) (*Nakanishi, 2022*), in this case, the two domains found in AGOs from mango might be a product of insertions in the middle of a unique PIWI domain. In the latter case, our expression analysis shows that this gene has a stress-responsive role and is probably a functional protein. More analysis must be conducted to prove this.

On the other hand, only MiAGO4b lacks an N-terminal domain in its structure, including a reduced PIWI domain (approximately 49 aa) at the beginning of the sequence, compared to the over 100 aa of other AGOs. In this context, the full total size of MiAGO4b was 548 aa, relatively small compared to its closest homologs in *C. sinensis* (898–920 aa) (*Sabbione et al., 2019*) and MiAGO4a (886 aa). Interestingly, the ArgoMid domain was found only in seven AGO sequences. The inability to identify this domain in almost half of the mango AGO sequences might stem from the scanning conditions applied, including the software settings, or divergence from the traditionally acknowledged amino acid sequence. A genuine absence would be highly unexpected, given that crystal structures have demonstrated ArgoMid presence in crucial interactions with the Piwi domain and for recognizing and binding the 5′ terminal phosphate of the guide RNA (*Davis-Vogel et al., 2018*). On the other hand, the evolutive history of AGO proteins has been characterized by

several gene losses and duplications that contributed to the diversification of sequences at different taxonomical levels (*Zhang et al., 2015*; *Singh et al., 2015*). In this sense, we found that six pairs of mango AGOs underwent a segmental duplication, contributing to their expansion in this fruit.

## HYL1, SE, HEN1, HST, and HSP90 as key elements in miRNAs production

HYL1 and SE proteins are core components of the protein complex called Microprocessor and help DCL1 process the pri-miRNAs efficiently and accurately (*Yang et al., 2006*; *Fang & Spector, 2007*). The HST protein also processes pri-miRNA and interacts with DCL1 (*Cambiagno et al., 2021*). The interaction of these proteins with DCL1 is carried out through their specific domains. For instance, according to the phenotype observed in *hyl1* null mutants in Arabidopsis, both dsRBDs located at the N-terminal are indispensable and sufficient for pri-miRNA processing (*Wu et al., 2007*). For its part, SE interacts with DCL1 through both the N-terminus and C-terminus, containing the zinc finger domain, the last and most important due to their stimulatory activity of DCL1. Also, SE can bind to RNA with its N-terminal domain (*Iwata et al., 2013*). Because mango HYL1 and SE possess these domains, they are highly likely functional proteins. Thus, *Bollman et al. (2003)* reported that the first 107 amino acids of HST have an essential role in the HST-DCL1 interaction; however, we found no specific domain in that region, even using the P kinase domain as a guide since it is less than 15 aa away from it. The authors have hypothesized that the mutant *hst* protein, lacking the first aa in the N-terminus, may not be able to interact with RAN1, leading to the formation of closed, ring-like structures that fail to bind pre-miRNAs, just like in animals (*Cambiagno et al., 2021*). Thus, functional analyses of MiHST are needed to prove how it works.

Interestingly, HEN1, the plant small RNA methyltransferase that methylates the miRNA/miRNA* duplex (*Yu et al., 2005*), interacts physically with HYL1 but not with SE (*Baranauskė et al., 2015*). *Huang et al. (2009)* determined the crystal structure of full-length Arabidopsis HEN1, finding multiple domains from the N- to the C-terminus: the protein harbors two dsRBDs, including a La-motif-containing domain, a FK506-binding protein-like domain, and a methyltransferase domain, very similar to what we found in mango. Functional analysis in Arabidopsis using different versions of the truncated protein revealed that the N-terminus is responsible for the RNA binding and the C-terminus for methyl transfer (*Vilkaitis, Plotnikova & Klimašauskas, 2010*; *Baranauskė et al., 2015*).

In addition, it has been shown that the conformation of AGO1 is modified to accommodate the process of RISC assembly, which also needs to be facilitated by many cofactors, such as HSP90, which is necessary for the export of miRNA from the nucleus to the cytoplasm (*Bologna et al., 2018*). The phylogeny of HSP90 has been more broadly studied than other accessory molecules in small RNA pathways. In different studies, the distribution of HSP90 protein sequences tends to form two major groups, group I and group II, based on classification criteria as in Arabidopsis (*Krishna & Gloor, 2001*). Such is the case of two studies analyzing the phylogenetic relationships between *Populus* (*Zhang*

*et al., 2013*) and carnation (*Xue et al., 2023*), where subsequent subgroups locate *O. sativa*, Arabidopsis, and fruit plants in different clades, consistent with our findings. Among all proteins analyzed here, HSP90 is the most conserved in its sequence. This could explain their close phylogenetic relationship even with *P. patens* and *S. moellendorffii*.

## miRNAs biogenesis pathway proteins contain stress-responsive *cis*-acting elements

The mango protein sequences showed common *cis*-acting regulatory elements in their promoter region. Identifying *cis*-regulatory DNA sequences is key to elucidating how genes coordinate responses to developmental and environmental cues (*Schmitz, Grotewold & Stam, 2022*). Furthermore, several stress-responsive elements in the promoter sequences of miRNA biogenesis-related genes were found. For example, MYB, MYC, and ABRE are core sequences for transcription factor binding mediated by abscisic acid (ABA), or DRE1, related to ethylene response in conditions usually associated with heat, such as dehydration and salinity (*Ijaz et al., 2020*). Previous publications about AGO and DCL recognize stress and defense response, plant growth and development, and light-responsive elements as the major groups of elements found in these genes (*Mishra et al., 2023*; *Jing et al., 2023*; *Podder et al., 2023*; *Ahmed et al., 2021*). Due to the close evolutionary history of mango and other fruit species, the elements in *C. sinensis* coincide with those in mango (*Mosharaf et al., 2020*).

## The QHWT modified the expression of miRNA's biogenesis genes

Gene expression was evaluated in mesocarp tissue from fruits subjected to QHWT. Curiously, the expression of the sRNAs biogenesis-related genes evaluated here is barely affected under control conditions. However, this behavior changed because of the heat stress, indicating an effect of the QHWT. Interestingly, *MiDCL1* and *MiAGO1* were affected similarly, while *MiDCL3* and *MiAGO6* also showed similar changes. These results could be related to the specific function of the proteins encoded in each gene. For example, DCL1 carries out the miRNA biogenesis, producing 21-nt sRNA (*Kurihara & Watanabe, 2004*), and AGO1 is the main effector protein implicated in the miRNA pathway (*Borges & Martienssen, 2015*).

On the other hand, DCL3 produces siRNAs of 24 nt in length (*Blevins et al., 2015*; *Liu et al., 2020*) related to transcriptional gene silencing (TGS); AGO4 and AGO6 are also involved in TGS. All these evaluated genes were induced at 6 and 24 h. Supporting this data, the *MiHST* reached their maximum level expression at 3 and 6 h post-treatment, just like *MiDCL1* and *MiAGO1b*. *MiHEN1*, for its part, was highly induced at 6 h, and its level was almost basal at 24 h, coinciding with repression of MiDCL1, probably because, at this time, miRNAs were no longer produced in response to heat stress.

Based on functional analysis in Arabidopsis, miR156 is upregulated under heat stress, while, in turn, its biogenesis requires DCL1 activity (*Kim et al., 2012*; *Stief et al., 2014*). Regarding this, increasing *DCL1* expression under heat stress conditions has been confirmed in tomato (*Bai et al., 2012*) and cucumber (*Gan et al., 2017*) after 3 h at 40 °C in both experiments, coinciding with the highest expression we found 3 h after application of

the QHWT. In the same studies, gene expression of *DCL3* in tomatoes was not different from normal conditions, contrary to cucumber, where *DCL3* showed an increase of transcripts predominantly in stem and flower. Depending on the organism and the tissue, this could suggest some specificity of the heat stress response.

On the other hand, a study in maize seedlings exposed to 40 °C found that 10 of the 17 *AGO* genes in the genome were upregulated by that treatment at 1 h, followed by a decrease at 2, 4, and 12 h (*Zhai et al., 2019*). In a different study, using a 4-month-old apple seedling, gene expression of *AGO1, 4,* and *6* were induced after 6 h under 37 °C conditions, *AGO1* being the highest induced, up to 13 times compared to others (*Xu et al., 2016a*). Similar observations were made on *AGOs* from cucumber, where the expression of the same orthologs was increased, especially in stems and flowers (*Gan et al., 2017*). Considering all this evidence, it seems that each AGO has its function regulating gene expression in different periods under heat stress, whereas *MiAGO1b* has an earlier response, *MiAGO6* has a briefer induction, and *MiAGO4a* has a more delayed but lasting effect after stress.

Regarding HSP90, in addition to its role in the biogenesis of miRNAs, these proteins are involved in response to heat stress. Studies in Arabidopsis show that an HSP90 negatively inhibited heat shock factors (HSF) without heat stress. Still, this role is temporarily suspended under heat stress, so HSF is active (*Yamada & Nishimura, 2008*). As other authors mention, reduced expression of the *HSP90* under heat stress may indicate a reduced role in inhibiting the HSF, leading to the activation of heat-responsive genes and heat acclimation (*Rosic et al., 2011*). Our findings suggest that transcripts of *HSP90* were induced before the other genes we analyzed, supporting the role of HSP90 in responding to heat stress conditions. Curiously, transcript levels are not induced from the first 60 min after QHWT exposure. But interestingly, their expression patterns resemble those of *MiDCL1* and *MiAGO1b*. This could suggest that the *MiHSP90* evaluated here relates more to miRNA biogenesis than heat response.

## CONCLUSIONS

Certain stress conditions may cause changes in small RNA molecules in *Mangifera indica*. However, more experiments are needed to confirm this theory. The presence of stress-responsive *cis*-elements in the promoters of miRNA biogenesis genes suggests this could be the case. This research provides a foundation for further exploration of small RNA molecules in mangoes and their potential application in postharvest management. This technology could significantly improve the shelf life, nutritional value, and quality of mangoes.

## ACKNOWLEDGEMENTS

The authors thank Célida Isabel Martínez Rodríguez and Rosalba Contreras Martínez for technical support.

### Funding

This work was supported by institutional funds from CIAD and IPICYT. Andrés G. López-Virgen and Lourdes K. Ulloa-Llanes received graduate fellowships, and Mitzuko Dautt-Castro received a Postdoctoral scholarship, all funded by CONAHCyT Mexico. The funders had no role in study design, data collection and analysis, decision to publish, or preparation of the manuscript.

### Grant Disclosures

The following grant information was disclosed by the authors:
Institutional funds from CIAD and IPICYT.
CONAHCyT Mexico.

### Competing Interests

Rogerio R. Sotelo-Mundo, PhD. is an Academic Editor for PeerJ.

### Author Contributions

- Andrés G. López-Virgen conceived and designed the experiments, performed the experiments, analyzed the data, prepared figures and/or tables, authored or reviewed drafts of the article, and approved the final draft.
- Mitzuko Dautt-Castro conceived and designed the experiments, performed the experiments, analyzed the data, prepared figures and/or tables, authored or reviewed drafts of the article, and approved the final draft.
- Lourdes K. Ulloa-Llanes performed the experiments, analyzed the data, prepared figures and/or tables, authored or reviewed drafts of the article, and approved the final draft.
- Sergio Casas-Flores conceived and designed the experiments, analyzed the data, authored or reviewed drafts of the article, and approved the final draft.
- Carmen A. Contreras-Vergara performed the experiments, authored or reviewed drafts of the article, and approved the final draft.
- Miguel A. Hernández-Oñate analyzed the data, authored or reviewed drafts of the article, and approved the final draft.
- Rogerio R. Sotelo-Mundo performed the experiments, analyzed the data, prepared figures and/or tables, authored or reviewed drafts of the article, and approved the final draft.
- Rosabel Vélez-de la Rocha performed the experiments, authored or reviewed drafts of the article, and approved the final draft.
- Maria A. Islas-Osuna conceived and designed the experiments, analyzed the data, prepared figures and/or tables, authored or reviewed drafts of the article, and approved the final draft.

### Field Study Permissions

The following information was supplied relating to field study approvals (i.e., approving body and any reference numbers):

Fruits were collected at a commercial orchard. Engineer Ismael Díaz Murillo from "Diazteca" orchard gave us the mango fruit used in the experiments.

## Data Availability

The sequences are public: https://mangobase.org/easy_gdb/index.php:
Manin16g013400.1, Manin03g001700.1, Manin03g001710.1, Manin05g003940.1,
Manin05g000540.1, Manin00g009890.1, Manin17g007920.1, Manin13g011150.1,
Manin18g011630.1, Manin01g005840.1, Manin03g006020.1, Manin03g010820.1,
Manin12g007340.1, Manin09g012380.1, Manin16g006430.1, Manin02g009270.1,
Manin16g005400.1, Manin07g002280.1, Manin18g010850.1, Manin03g011860.1,
Manin01g011660.1, Manin14g005250.1, Manin15g005480.1, Manin15g010550.1,
Manin16g001590.1, Manin18g009710.1, Manin01g005900.1, Manin03g005950.1,
Manin01g011860.1, Manin04g001850.1, Manin10g008510.1, Manin19g016320.1,
Manin20g008060.1.

The qRT-PCR data are available in the Supplemental Files for each of the genes, for the untreated (CONTROL) and treated fruits (QHWT).

## Supplemental Information

Supplemental information for this article can be found online at http://dx.doi.org/10.7717/peerj.17737#supplemental-information.

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
