# Peer review of "Genome-wide identification of gene families related to miRNA biogenesis in Mangifera indica L. and their possible role during heat stress"

_PeerJ, doi:10.7717/peerj.17737_

## Round 0.1 · original submission · Major Revisions

The manuscript "Genome-wide identification of Dicer, HEN, HASTY, and Argonaute gene families related to the miRNA biogenesis in Mangifera indica L. and their possible role during the heat stress" by Andrés G López-Virgen and colleagues needs major revisions. Its primary issue is the poor quality of English, which significantly affects readability. This must be addressed thoroughly. Additionally, the scientific content, particularly the rationale behind gene selection and the exclusion of important cofactors, needs clearer argumentation and evidence. The manuscript structure is adequate, but specific aspects, like data presentation and experimental design, require refinement. Overall, a comprehensive review and implementation of the provided suggestions are crucial for the manuscript's improvement and potential acceptance for publication. Please, reply to all observations made by the referees.

**Language Note:** The Academic Editor has identified that the English language must be improved. PeerJ can provide language editing services - please contact us at [email protected] for pricing (be sure to provide your manuscript number and title). Alternatively, you should make your own arrangements to improve the language quality and provide details in your response letter. – PeerJ Staff

Reviewer 1 ·

Basic reporting

a. Clear, unambiguous, professional English language used throughout.
The English language needs substantial improvement. Please consider asking a colleague who is proficient in English and familiar with the subject
matter to review your manuscript, or contacting a professional
editing service.
There are several instances where the intent is unclear e.g. 187-189, or where informal language is used e.g. ‘thrilling’ at Line 527, ‘suffered’ at Line 385. There are also several instances of grammatical and spelling errors. Some examples are:
- Title: remove both ‘the’ from the title as unnecessary
- Lines 64-66: The miRNA/miRNA7 duplexes are subsequently transported from the nucleus to the cytoplasm through the CRM1 (EXPO1)/NES-dependent manner via TREX-2, and nucleoporin protein (NUP1) facilitated pathway
Should be ‘through the CRM1(EXPO1)/NES-dependent pathway’ OR ‘in a CRM1(EXPO1)/NES-dependent manner’
- Lines 67-69: Once in the cytoplasm, the duplex requires the activity of the ATP-dependent chaperone activity of Hsc70/Hsp90 to get loaded onto the AGO1 by the opening of the nucleic acid-binding channel of the AGO protein
Repetition of the word activity.
- Line 85: Change specie to species
- Lines 62-64: Suggest adding ‘The miRNA duplex’ to make it clear it’s the miRNA being methylated

b. Intro & background to show context. Literature well referenced & relevant.
The introduction and background give enough information to understand the article.

c. Structure conforms to PeerJ standards, discipline norm, or improved for clarity.
The structure conforms to the standards. However, the results section is very basic and mostly just reiterates what is in the table/figure without giving any context or interpretation. This section needs significant improvement. Please see the following comments for specific examples:
Lines 187-190: Did you just select the top BLAST hits as the potential orthologs? How did you name each gene within a gene family? You probably should present the gene tree results here to validate your assignment of each mango gene.
Lines 191-205: What is the purpose of this information? If you are going to repeat the information presented in the table at least compare it to other species e.g. MiDCL proteins had a similar length to Arabidopsis DCL??
Lines 205-207: Why did you choose MiHSP90? Was it a random selection or did some of the properties of MiHSP90 lead to you selecting it?
Lines 209-214: Again, what is the purpose of presenting the number of introns? The structure of the DCL genes do not seem consistent based on these figures. Perhaps you could focus on the differences, rather than just stating the number of introns?
Lines 217-234: What are all these domains? Which ones are important for which protein function? Instead of just stating what domains are present can you please put the results in context e.g. X domain is important for X function. All DCL genes except DCL2 contain this domain suggesting that DCL2 may not be functional. (see Lines 256-267 where you have done this better).
Lines 271-274: Not sure this information is useful.
Lines 294-308: This could be explained so much better. Please just describe the overall pattern of expression rather than each individual gene. What is this data showing us?

d. Figures are relevant, high quality, well labelled & described.
The figures are mostly relevant to the article, well labelled and described. Figure 1 is blurry and Figure 3 some of the text is too small to read (consider rearranging the figures to make the font a similar size in all 5 sub-figures.
Specific comments:
Figure 4: Not sure this is the best way to present this information. It is difficult to compare #s between so many different genes.
Figure 7: consider providing supplementary figures of this data showing the average and standard errors for each gene and sample. Heat maps are useful for observing general trends, but lack information regarding variability between biological replicates. Also, I was unsure from the methods whether control were harvested at each time point, or just at time 0, can you please clarify this.

e. Raw data supplied (see PeerJ policy).
Raw data was supplied. However, can you please provide appropriate labelling of the samples in the qPCR output?

Experimental design

a. Original primary research within Scope of the journal.
The research article is within the scope of the journal. However, I would question whether the standard of this paper is high enough.

b. Research question well defined, relevant & meaningful. It is stated how the research fills an identified knowledge gap.
The article was aiming to identify genes involved in miRNA biogenesis and determine their role in heat stress response.

c. Rigorous investigation performed to a high technical & ethical standard.
The article is a very basic investigation to identify genes in mango and look at their expression after treatment. The description of the results and interpretation is lacking (as described elsewhere).

d. Methods described with sufficient detail & information to replicate.
Some methods are lacking in detail. Please provide more detail on the following:
Line 113: What settings of CLUSTAL-W were used? What do you mean by overlapping genes?
Line 128: Maximum likelihood is the preferred method to construct gene trees. Note, you are constructing a gene tree not a phylogenetic tree. You might need to include an outgroup in your alignments.
Line 143: The server link here does not seem to exist anymore. Can you please provide further information about this server?

Validity of the findings

a. Impact and novelty not assessed. Meaningful replication encouraged where rationale & benefit to literature is clearly stated.
The authors replicated the qPCR experiment with three biological replicates which is the minimum required. Otherwise the rest of the article was based on bioinformatics.

b. All underlying data have been provided; they are robust, statistically sound, & controlled.
Underlying data provided. Although as stated earlier, a supplementary figure for Figure 7 to show the variation between biological replicates would be ideal.

c. Conclusions are well stated, linked to original research question & limited to supporting results.
The second half of the conclusions (Line 528-535) should be removed as these are not conclusions.
Specific comments on the discussion:
Lines 366-368: Why is this interesting?
Lines 380-382: Can you please explain this further? How do P. patens and S. moellendorffii amplify the divergence?
Lines 385-386: Only 3-4 pairs of MiAGO may have undergone segmental duplication in mango. The rest appear to have occurred prior to mango diverging from other species. Also, consider using a different word than ‘suffered’ e.g. ‘underwent’

Additional comments

Some abbreviations are missing e.g. Lines 56-63 DCL, AGO, MIR, HST, HEN1, EXPO1, NES

Reviewer 2 ·

Basic reporting

- The English language should be revised, there are numerous typos and grammatical issues that need correction. Some examples are outlined in point 8 of General comments.
- The introduction needs to be improved; the details are indicated in point 7 of General comments.
- In general, the literature is ok, but there are some recent articles and/or reviews that should be read and cited.
- The structure of the manuscript is ok.
- Some figures should be improved; the details are indicated in point 4.
- Raw data supplied is correct.

Experimental design

In this interesting study, the authors identified the Arabidopsis thaliana and tomato DCL, AGO, HEN1, HST1 and HSP90 orthologs in mango (Mangifera indica L.) genome, obtaining 5 MiDCL, 13 MiAGO, 1 MiHEN1, 1 MiHST and 5 MiHSP90 gene sequences. Gene structure and conserved domains analysis were performed. In general, the identified proteins contain the major and most important domains for their functions in sRNA pathways, as expected. Phylogenetic analysis showed that the identified mango proteins were correctly grouped into previously reported clades, including the DCL and AGO family proteins. In addition, the promoters of all these genes contained many abiotic stress-responsive cis-regulatory elements, most of them related to heat and hypoxia conditions. In line with this, expression analysis in mango mesocarp exposed to QHWT were performed, showing that the selected transcripts to be measured (MiDCL1, MiDCL3, MiAGO1b, MiAGO4a, MiAGO16 -or MiAGO6?-, MiHEN1, MiHST, and MiHSP90) are differentially expressed under the treatment conditions.

Overall, the experiments are well conducted, but some key aspects need to be further investigated in order to draw the conclusions that the authors claim in the text and title.

Validity of the findings

The identification of DCLs, AGOs, and HEN1 genes in the Mangifera indica L. genome and their description are the strengths of this study. However, there is not enough argumentation about the selection of these genes and why the authors have excluded from the analysis very important and well described cofactors such as HYL1 and SE, but including HASTY, which is one among numerous proteins that acts in the recruitment of microprocessor complex to the MIRNA loci promoting co-transcriptional processing. In the case of HSP90, its importance is similar to that of SQN, another protein excluded from this study that acts in the RISC assembly process. The authors also present data showing differential expression of some of the identified genes, measured by qRT-PCR, in mango fruits exposed to hydrothermal treatment at different times, and conclude that these results suggest a miRNA (and other small RNAs) production in response to the treatment. This conclusion is not sufficiently supported by the data presented, because to draw this conclusion it is necessary to demonstrate at least the production of some stress-related miRNAs by measuring their levels and the repression of their target transcripts. The main issues and the experimental approaches to address them are outlined below, along with other minor comments.

Major points

1. The authors should include in this study the well described miRNA pathway co-factors HYL1, SE, and also SQN. At least the first 2, as they are key proteins in the microprocessor complex. HYL1 is involved in almost all steps of miRNA biogenesis, since MIR genes transcription to mature duplex loading into AGO1. Also, the authors should include a MiHSP90 orthologue to an A. thaliana nuclear HSP90, since it was demonstrated that it is involved in miRNA nuclear loading into AGO1.

2. The authors should measure the sRNAs levels in mango fruits exposed to QHWT treatment at different times, as they mentioned in the conclusions, to analyze the different types of sRNA produce during the treatment (miRNAs, 21-nt sRNA, 24-nt sRNAs). If difficult, at least measure the levels of some stress-responsive miRNAs by Northern blot or stem-loop qRT-PCR. Also, measure the levels of their transcript targets by qRT-PCR.

Additional comments

Minor points and comments

3. The authors should move Figure 1 from the main text to the Supplementary Data and include as Figure 1 a figure showing the alignments of the A. thaliana, tomato, and mango proteins, as this would be more informative. It would even be interesting to plot the domains identified and shown in Figure 2 in the alignment representation.

4. For phylogenetic analysis, the authors should use a most robust method such as Maximum Likelihood (ML) rather than Neighbor Joining (NJ), since the latter is not often used.

5. In Figure 3, panels A and B (DCL and AGO phylogenetic trees) are too small. The authors should design the figures to make the trees as easy to read as possible.

6. In Figure 7, I recommend the authors change the heat map to the bar graphs in the supplemental data, as the latter is easier to visualize and analyze.

7. The introduction needs to be improved. Some points are contradictory between the introduction and the discussion/conclusion. For instance, in lines 64-66, the mechanism of miRNA duplex transport from the nucleus to the cytoplasm is poorly described, without mentioning the nuclear miRNA loading into AGO1 and the involvement of HSP90 in this process, but only the loading in the cytoplasm through AGO1 and HSP90. Nevertheless, in lines 435-437 of the discussion, the process of miRNA export from the nucleus to the cytoplasm is described as a process that depends on the interaction of AGO1 and HSP90, which leads to confusion. Also, HASTY, one of the proteins selected for the study, is mentioned but not described in the introduction.

8. Finally, there are numerous typos and grammatical issues that need correction. For example:

• Line 31: HUA ENHANCER1 (HEN1)
• Line 63: “Hua HEN1” is wrong. The name of the protein is HUA ENHANCER1 (HEN1).
• Line 94: The reference (Barciszewska-Pacak et al., 2015) does not appear in Bibliography.
• Line 294: (hours 0 and 1). I guess that “0” is wrong and the correct hour is “0.5”. This error must be corrected in figures and other lines of the text.
• Line 394: “MiAGO6” should be MiAGO16? This error must be corrected in figures and other lines of the text.
• Line 419: “HSTY” should be “HASTY” or “HST”. “least” should be “last”?
• Lines 419-421: This sentence reads oddly.
• Lines 448-451: This sentence is confusing.
• Line 607: “Bailong Z.” should be “Zang B.”.

---

## Round 0.2 · Minor Revisions

Both reviewers note that the authors have improved manuscript readability, but they also highlight areas still requiring further revision. All comments from reviewers 1 and 2 should be addressed. This includes solving minor spelling and grammatical errors, confirming and correcting labeling in supplementary files, providing legends for all supplementary materials, and correcting errors in figures. Additionally, clarifications in the experimental design and the validity of findings are necessary. By addressing these points, the manuscript will be improved accordingly.

Reviewer 1 ·

Basic reporting

The language used throughout is much improved. A few very minor spelling/grammatical errors remain (examples listed below in Specific comments).

The introduction reads well now.

Thank you for improving the written results section. It is much easier to read and interpret now.

The authors have addressed the previous comments regarding the raw data. However, please check all samples are labelled correctly in ‘Relative Gene Expression Raw Data and Analysis.xlsx’ file. For example, I believe sample 4 should read Replicate 1, not Replicate 2. While sample 9 is missing the Replicate # (should be Replicate 3 I believe).
Also, the Table S1 file contains an old copy of Figure 3?
Finally, there do not appear to be any figure or table legends for any of the supplementary files, apart from Table S1. Please confirm if there will be legends supplied for the supplementary files or not.

Specific comments:
Please italicize gene names throughout the manuscript (mostly missing in the introduction and results e.g. MIR genes in Lines 70, 76. Lines 88, 96, 97, 222).
Or if you are referring to the protein, then use protein instead of gene e.g. Line 222 is referring to amino acid sequences, so change to ‘all MiDCL proteins’

Line 56: Which type of fruit fly are you referring to? Asian fruit fly? Or if general, could use ‘exotic fruit fly species’?

Line 69: DCL is an acronym for DICER-LIKE

Line 70: you haven’t used the MIR acronym yet, so please spell out in full.

Line 70: change to ‘these are a unique type

Line 72: Change DLC1 to DCL1. Check this is correct throughout manuscript.

Line 81: Remove ‘it’ after (miRSIC)

Line 136: Lower case k for kb

Line 369: Change ‘leves’ to ‘level’

Experimental design

Methods are adequately described, however the following could be provided to improve the ability to replicate the results:

Line 105: Which genome did you search? Tommy Atkins, Alphonso, or both?

For replication, it would be useful to have a supplementary file listing the NCBI ID for each of the genes used for BLASTp against the mango genome. (From lines 125-127).

Other aspects of the experimental design have been adequately addressed by the authors.

Validity of the findings

The only comment that I still have issue with is:
Lines 385-386: Only 3-4 pairs of MiAGO may have undergone segmental duplication in mango. The rest appear to have occurred prior to mango diverging from other species. Also, consider using a different word than ‘suffered’ e.g. ‘underwent’
R= Dear reviewer, we need some clarification on this comment. Bally et al 2021 reported: "A Ks distribution analysis of the coding sequences revealed that the mango genome had a whole genome duplication (WGD) dated at 65 MYA (Ks = 0.270). It is not shared with Pistachia vera from which it diverged 61 MYA (Ks=0.200) in agreement with other phylogenetic studies". According to our calculations, the diverging time of all the AGO genes is less than 65 MYA (Table S2). For that reason, we reported that the 6 pairs had undergone segmental duplication.

Response: This comment was based on interpreting both the results from Figures 2 and 5. According to Figure 2, MiAGOPNH1 and MiAGO10 both have closely related copies in both citrus and tomato which seems to imply that the divergence of these genes occurred prior to when the species diverged. The other pairs of genes are more likely to be segmental duplications occurring after mango diverged from other species. However, after more closely reviewing these figures again, I noticed that the labelling in both Figures is not consistent. In Figure 2, MiAGO4a is in the same clade as MiAGO4b and MiAGO6, but in Figure 5 MiAGO4a duplicated partner is labelled MiAGO16. Is the duplicated partner of MiAGO4a, MiAGO16, MiAGO6 or MiAG4b? Please confirm labelling of this gene. If the duplicated partner of MiAGO4a is MiAGO4b, then I agree that it is likely a segmental duplication as they are very closely related as per Figure 2. However, if the duplicated partner of MiAGO4a is MiAGO6 then I do not agree that the segmental duplication occurred after their divergence from other species as there are copies of both of these genes present in multiple species that are more similar to the other species than each other. Note, the labelling is also incorrect in Figure 4, where MiAGO16 is labelled on chromosome 12.

Additional comments

Thank you to the authors for addressing all of the reviewers’ comments and significantly improving the manuscript.

Reviewer 2 ·

Basic reporting

- The English language is ok, but there are some typo issues that should be revised:
• Abstract: “DLC” should be “DCL”
• Line 34: “DLC” should be “DCL”
• Line 273: “AGO4/6/8/9/16”, AGO16 should be AGO6. This should be changed in Figures 2, 4 and 5.
• Line 400: “onlyin” should be “only in”.
- Some figures should be improved:
• Figure 1 is blurry, the numbers of the aminoacid positions are not clear.
• Figure 4 some text is too small.
• Figure 2, 4 and 5: The error “AGO16” has not been corrected with respect to the last version.

I have no further comments, as the authors have addressed my concerns.

Experimental design

I have no further comments, as the authors have addressed my concerns.

Validity of the findings

I have no further comments, although there are some minor points and comments that the authors should address (see point 4).

Additional comments

Minor points and comments

1. Line 80: The selection of the duplex strand occurs once it is loaded into AGO1.
2. Line 257-259: I do not think it is appropriate to compare this protein to the mammalian protein because it has been shown that the function of HST in the miRNA pathway is not the same as that of Exportin5.
3. Line 355: DRB1 is HYL1. This protein is referred to by these two names, although HYL1 is the most commonly used.
4. Line 357: What does “two gene models” mean? Why do you use the “TAIR development paper” as a reference of this? I mean, the paper doesn’t study the gene models of HYL1.
5. Line 361: In Arabidopsis the function of HEN1 is well described, and there are also some interesting advances in the description of HST function.
6. Line 390: MiDCL2a and b have one RNase III domain. What would this mean for these proteins? Wouldn't they be functional in the sRNA pathways?
7. Line 469: It is better to reference the Bologna et al 2018 paper rather than a review.

---

## Round 0.3 · accepted · Accept

Authors have addressed all of the reviewers' comments and it is ready for publication.

Reviewer 1 ·

Basic reporting

The authors have addressed all the reviewer's comments and I am happy for the paper to be accepted without any further revisions.

Experimental design

The authors have addressed all the reviewer's comments and I am happy for the paper to be accepted without any further revisions.

Validity of the findings

The authors have addressed all the reviewer's comments and I am happy for the paper to be accepted without any further revisions.

Reviewer 2 ·

Basic reporting

I have no comments to make to the authors, as they have addressed all my concerns.

Experimental design

I have no comments to make to the authors, as they have addressed all my concerns.

Validity of the findings

I have no comments to make to the authors, as they have addressed all my concerns.

Additional comments

Thank you the authors for their attention to the reviewers' comments, the improvement of the manuscript has been noticeable.